# Learning Transferable Graph Exploration

**Hanjun Dai**[††*], **Yujia Li**[§], **Chenglong Wang**[‡], **Rishabh Singh**[†], **Po-Sen Huang**[§], **Pushmeet Kohli**[§]

[†] Georgia Institute of Technology
[†] Google Brain, {hadai, rising}@google.com
[‡] University of Washington, clwang@cs.washington.edu
[§] DeepMind, {yujiali, posenhuang, pushmeet}@google.com

## Abstract

This paper considers the problem of efficient exploration of unseen environments, a key challenge in AI. We propose a 'learning to explore' framework where we learn a policy from a distribution of environments. At test time, presented with an unseen environment from the same distribution, the policy aims to generalize the exploration strategy to visit the maximum number of unique states in a limited number of steps. We particularly focus on environments with graph-structured state-spaces that are encountered in many important real-world applications like software testing and map building. We formulate this task as a reinforcement learning problem where the 'exploration' agent is rewarded for transitioning to previously unseen environment states and employ a graph-structured memory to encode the agent's past trajectory. Experimental results demonstrate that our approach is extremely effective for exploration of spatial maps; and when applied on the challenging problems of coverage-guided software-testing of domain-specific programs and real-world mobile applications, it outperforms methods that have been hand-engineered by human experts.

## 1 Introduction

Exploration is a fundamental problem in AI; appearing in the context of reinforcement learning as a surrogate for the underlying target task [1, 2, 3] or to balance exploration and exploitation [4]. In this paper, we consider a *coverage* variant of the exploration problem where given a (possibly unknown) environment, the goal is to reach as many distinct states as possible, within a given interaction budget.

The above-mentioned state-space coverage exploration problem appears in many important real-world applications like software testing and map building which we consider in this paper. The goal of software testing is to find as many potential bugs as possible with carefully designed or generated test inputs. To quantify the effectiveness of program exploration, program coverage (e.g. number of branches of code triggered by the inputs) is typically used as a surrogate objective [5]. One popular automated testing technique is fuzzing, which tries to maximize code coverage via randomly generated inputs [6]. In active map building, a robot needs to construct the map for an unknown environment while also keeping track of its locations [7]. The more locations one can visit, the better the map reconstruction could be. Most of these problems have limited budget (e.g. limited time or simulation trials), thus having a good exploration strategy is important.

A crucial challenge for these problems is of generalization to unseen environments. Take software testing as an example, in most traditional fuzzing methods, the fuzzing procedure will start from scratch for a new program, where the knowledge about the previously tested programs is not utilized. Different programs may share common design patterns and semantics, which could be exploited during exploration. Motivated by this problem, this paper proposes a 'learning to explore' framework

where we learn a policy from a distribution of environments with the aim of achieving *transferable* exploration efficiency. At test time, presented with an unseen environment from the same distribution, the policy aims to generalize the exploration strategy to visit the maximum number of unique states in a limited number of steps.

We formulate the state-space coverage problem using Reinforcement Learning (RL). The reward mechanism of the corresponding Markov Decision Process (MDP) is non-stationary as it changes drastically as the episode proceeds. In particular, visiting an unobserved state will be rewarded, but visiting it more than once is a waste of exploration budget. In other words, the environment is always expecting something new from the agent.

States in many such exploration problems are typically structured. For example, programs have syntactic or semantic structures [8], and efficiently covering the program statements require reasoning about the graph structure. The states in a generic RL environment may also form a graph with edges indicating reachability. To utilize the structure of these environments, we augment our RL agent with a graph neural network (GNNs) [9] to encode and represent the graph structured states. This model gives our agent the ability to generalize across problem instances (environments). We also use a graph structured external memory to capture the interaction history of the agent with the environment. Adding this information to the agent's state, allows us to handle the non-stationarity challenge of the coverage exploration problem. The key contributions of this paper can be summarized as:

- We propose a new problem framework of exploration in graph structured spaces for several important applications.

- We propose to use GNNs for modeling graph-structured states, and model the exploration history as a sequence of evolving graphs. The modeling of a sequence of evolving graphs in particular is as far as we know the first such attempt in the learning and program testing literature.

- We successfully apply the graph exploration agent on a range of challenging problems, from exploring synthetic 2D mazes, to generating inputs for software testing, and finally testing real-world Android apps. Experimental evaluation shows that our approach is comparable or better in terms of exploration efficiency than strong baselines such as heuristics designed by human experts, and symbolic execution using the Z3 SMT (satisfiability modulo theories) solver [10].

## 2   Problem Formulation

We consider two different exploration settings. The first setting concerns *exploration in an unknown environment*, where the agent observes a graph at each step, with each node corresponding to a visited unique *environment state*, and each edge corresponding to an experienced transition. In this setting, the graph grows in size during an episode, and the agent maximizes the speed of this growth.

The second setting is about *exploration in a known but complex environment*, and is motivated by program testing. In this setting, we have access to the program source code and thus also its graph structure, where the nodes in the graph correspond to the program branches and edges correspond to the syntactic and semantic relationship between branches. The challenge here is to reason about and understand the graph structure, and come up with the right actions to increase graph coverage. Each action corresponds to a test input which resides in a huge action space and has rich structures. Finding such valuable inputs is highly non-trivial in automated testing literature [5, 11, 12, 13, 14], because of challenges in modeling complex program semantics for precise logical reasoning.

We formalize both settings with the same formulation. At each step $t$, the agent observes a graph $G_{t-1} = (V_{t-1}, E_{t-1})$ and a coverage mask $c_{t-1} : V_{t-1} \mapsto \{0, 1\}$, indicating which nodes have been covered in the exploration process so far. The agent generates an action $x_t$, the environment takes this action and returns a new graph $G_t = (V_t, E_t)$ with a new $c_t$. In the first setting above, the coverage mask $c_t$ is 1 for any node $v \in V_t$ as the graph only contains visited nodes. While in the second setting, the graph $G_t$ is constant from step to step, and the coverage mask $c_t(v) = 1$ if $v$ is covered in the past by some actions and 0 otherwise. We set the initial observation for $t = 0$ to be $c_0$ mapping any node to 0, and in the first exploration setting $G_0$ to be an empty graph. The exploration process for a graph structured environment can be seen as a finite horizon Markov Decision Process (MDP), with the number of actions or steps $T$ being the budget for exploration.

**Action** The space for actions $x_t$ is problem specific. We used the letter $x$ instead of the more common letter $a$ to highlight that these actions are sometimes closer to the typical inputs to a neural network,

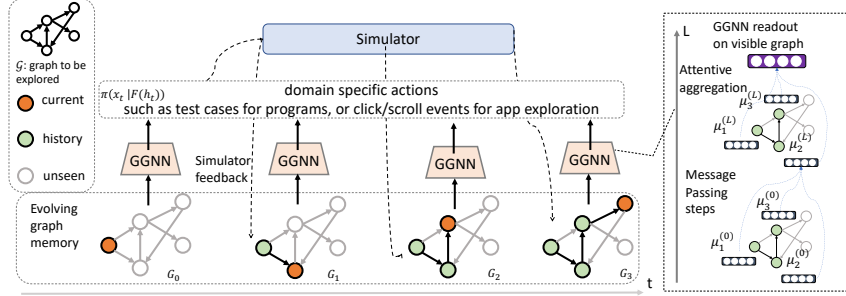

Figure 1: Overview of our meta exploration model for exploring a known but complicated graph structured environment. The GGNN [15] module captures the graph structures at each step, and the representations of each step are pooled together to form a representation of the exploration history.

which lives in an exponentially large space with rich structures, than to the more common fixed finite action spaces in typical RL environments. In particular, for testing programs, each action is a test input to the program, which can be text (sequences of characters) or images (2D array of characters).

Our task is to provide a sequence of $T$ actions $x_1, x_2, \ldots, x_T$ to maximize an exploration objective. An obvious choice is the number of unique nodes (environment states) covered, *i.e.* $\sum_{v \in V_T} c_T(v)$. To handle different graph sizes during training, we further normalize this objective by the maximum possible size of the graph $|\mathcal{V}|^2$, which is the number of nodes in the underlying full graph (for the second exploration setting this is the same as $|V_T|$). We therefore get the objective in Eq. (1).

$$\max_{\{x_1, x_2, \ldots, x_T\}} \sum_{v \in V_T} c_T(v)/|\mathcal{V}| \quad (1) \qquad r_t = \sum_{v \in V_t} c_t(v)/|\mathcal{V}| - \sum_{v \in V_{t-1}} c_{t-1}(v)/|\mathcal{V}|, \quad (2)$$

**Reward** Given the above objective, we can define the per-step reward $r_t$ as in Eq. (2). It is easy to verify that $\sum_{t=1}^{T} r_t = \sum_{v \in V_T} c_T(v)/|\mathcal{V}|$, *i.e.*, the cumulative reward of the MDP is the same as the objective in Eq. (1), as $\sum_{v \in V_0} c_0(v) = 0$. In this definition, the reward at time step $t$ is given to only the additional coverage introduced by the action $x_t$.

**State** Instead of feeding in only the observation $(G_t, c_t)$ at each step to the agent, we use an *agent state* representation that contains the full interaction history in the episode $h_t = \{(x_\tau, G_\tau, c_\tau)\}_{\tau=0}^{t-1}$, with $x_0 = \emptyset$. An agent policy maps each $h_t$ to an action $x_t$.

## 3 Model

**Overview of the Framework** We aim to learn an action policy $\pi(x|h_t; \theta_t)$ at each time step $t$, which is parameterized by $\theta_t$. The objective of this specific MDP is formulated as: $\max_{[\theta_1, \ldots, \theta_T]} \sum_{t=1}^{T} \mathbb{E}_{x_t \sim \pi(x|h_t; \theta_t)} r_t$. Note that, we could share $\theta$ across time steps and learn a single policy $\pi(x|h_t, \theta)$ for all $t$, but in a finite horizon MDP we found it beneficial to use different $\theta$s for different time steps $t$.

In this paper, we are not only interested in efficient exploration for a single graph structured environment, but also the *generalization* and *transferrability* of learned exploration strategies that can be used without fine-tuning or retraining on unseen graphs. More concretely, let $\mathcal{G}$ denote each graph structured environment, we are interested in the following *meta*-reinforcement learning problem:

$$\max_{[\theta_1, \ldots, \theta_T]} \mathbb{E}_{\mathcal{G} \sim \mathcal{D}} \left( \sum_{t=1}^{T} \mathbb{E}_{x_t^{(\mathcal{G})} \sim \pi(x|h_t^{(\mathcal{G})}; \theta_t)} r_t^{(\mathcal{G})} \right) \quad (3)$$

where $\mathcal{D}$ is the distribution of graph exploration problems we are interested in, and we share the parameters $\{\theta_t\}$ across graphs. After training, the learned policy can generalize to new graphs $\mathcal{G}' \sim \mathcal{D}$ from the same distribution, as the parameters are not tied to any particular $\mathcal{G}$.

**Graph Structured Agent and Exploration History**    The key to developing an agent that can learn to optimize Eq. (3) well is to have a model that can: 1) effectively exploit and represent the graph structure of the problem; and 2) encode and incorporate the history of exploration.

Fig. 1 shows an overview of the agent structure. Since the observations are graph structured in our formulation, we use a variant of the Graph Neural Network [9] to embed them into a continuous vector space. We implement a mapping $g : (G, c) \mapsto \mathbb{R}^d$ using a GNN. The mapping starts from initial node features $\mu_v^{(0)}$, which is problem specific, and can be e.g. the syntax information of a program branch, or app screen features. We also pad these features with one extra bit $c_t(v)$ to add in run-time coverage information. These representations are then updated through an iterative message passing process,

$$\mu_v^{(l+1)} = f(\mu_v^l, \{(e_{uv}, \mu_u^{(l)})\}_{u \in \mathcal{N}(v)}), \tag{4}$$

where $\mathcal{N}(v)$ is the neighbors of node $v$, $e_{uv}$ is the feature for edge $(u, v)$. This iterative process goes for $L$ iterations, aggregating information from $L$-hop neighborhoods. We use the parameterization of GGNN [15] to implement this update function $f(.)$. To get the graph representation $g(G, c)$, we aggregate node embeddings $\mu_v^{(L)}$ from the last message passing step through an attention-based weighted-sum following [15], which performs better than a simple sum empirically.

Capturing the exploration history is particularly important. Like many other similar problems in RL, the exploration reward is only consistent when taking the history into account, as repeatedly visiting a 'good' state can only be rewarded once. Here we treat the $h_t$ as the evolving graph structured memory for the history. The representation of the full history is obtained by aggregating the per-step representations. Formally, we structure the representation function $F$ for history $h_t$ as $F(h_t) = F([g_x(x_0), g(G_0, c_0)], ..., [g_x(x_{t-1}), g(G_{t-1}, c_{t-1})])$, where $g_x$ is an encoder for actions and $[\cdot]$ is the concatenation operator. The function $F$ can take a variety of forms, for example: 1) take the most recent element; or 2) auto-regressive aggregation across $t$ steps. We explored a few different settings for this, and obtained the best results with an auto-regressive $F$ (more in Appendix B.1).

The action policy $\pi(x_t|h_t) = \pi(x_t|F(h_t))$ conditioned on an encoding of the history $h_t$ is parameterized by a domain specific neural network. In program testing where the actions are the generated program inputs, $\pi$ is an RNN sequence decoder; while in other problems where we have a small finite set of available actions, an MLP is used instead.

**Learning**   To train this agent, we adopt the advantage actor critic algorithm [16], in the synchronized distributed setting. We use 32 distributed actors to collect on-policy trajectories in parallel, and aggregate them into a single machine to perform parameter update.

## 4   Experiments

In this section, we first illustrate the effectiveness of learning an exploration strategy on synthetic 2D mazes, and then study the problem of program testing (coverage guided fuzzing) through learning, where our model generates test cases for programs. Lastly we evaluate our algorithm on exploring both synthetic and real-world mobile Apps. We use GMETAEXP (Graph Meta-Exploration) to denote our proposed method. More details about experiment setup and more results are included in Appendix B.

To train our agent, we adopt the advantage actor critic algorithm [16] in the synchronized distributed setting. For the synthetic experiments where we have the data generator, we generate the training graph environments on the fly. During inference, the agent is deployed on unseen graphs and not allowed to fine-tune its parameters (zero-shot generalization).

The baselines we compare against fall into the following categories:

- **Random exploration**: which randomly picks an action at each step;
- **Heuristics**: including exploration heuristics like depth-first-search (DFS) or expert designed ones;
- **Exact Solver**: we compare with Z3 when testing programs. This is a state-of-the-art SMT solver that can find provably optimal solutions given enough computation time.
- **Fuzzer**: we compare with state-of-the-art fuzzing tools like AFL [3] and Neuzz for program testing.
- **RL baselines**: we also compare with RL models that use different amount of history information, or different history encoding models.

| DSL | # train | # valid | # test | Coverage |
|---|---|---|---|---|
| RobustFill | 1M | 1,000 | 1,000 | RegEx |
| Karel | 212,524 | 490 | 467 | Branches |

Table 1: DSL program dataset information.

| Method | Random | RandomDFS | GMETAEXP |
|---|---|---|---|
| Coverage | 33% | 54% | **72%** |

Table 2: Fraction of the mazes covered via different exploration methods.

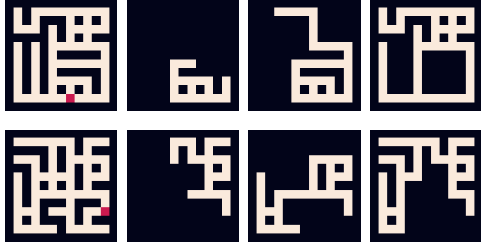

Full Maze    Random    RandDFS    GMETAEXP

Figure 2: Maze exploration visualizations. Note the mazes are 6x6 but the walls also take up 1 pixel in the visualizations. The start position is marked red in the first column.

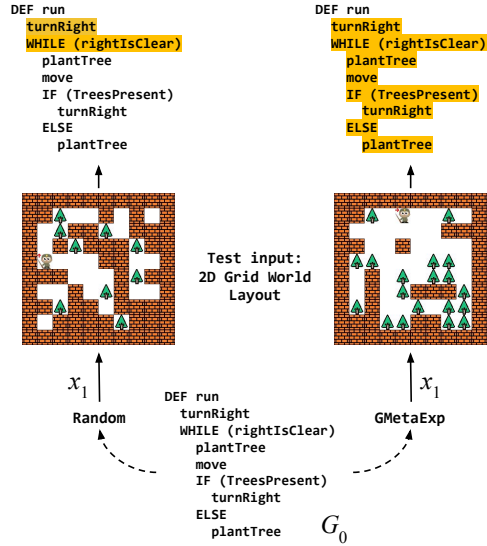

Figure 3: Test cases (2D grid world layouts) generated for Karel. Covered program branches are marked . The generated layout on the right by our model GMETAEXP covers all statements in the program, while the program exits after the first statement using the layout on the left.

## 4.1 Synthetic 2D Maze Exploration

We start with a simple exploration task in synthetic 2D mazes. The goal is to visit as much of a maze as possible within a fixed number of steps. This is inspired by applications like map building, where an agent explores an environment and builds a map using e.g. SLAM [7]. In this setup, the agent only observes a small neighborhood around its current location, and does not know the 2D coordinates of the grids it has visited. At each time step, it has at most 4 actions (corresponding to the 4 directions).

More concretely, we use the following practical protocol to setup this task:

- **Observation**: the observed $G_t$ contains the locations (nodes) and the connectivities (edges) for the part of the maze the agent has traversed up to time $t$, plus 1-hop vision for the current location.

- **Reward**: as defined in Eq. (2), a positive reward will only be received if a new location is visited;

- **Termination**: when the agent has visited all the nodes, or has used up the exploration budget $T$.

We train on random mazes of size $6 \times 6$, and test on 100 held-out mazes from the same distribution. The starting location is chosen randomly. We allow the agent to traverse for $T = 36$ steps, and report the average fraction of the maze grid locations covered on the 100 held-out mazes.

Table 2 shows the quantitative performance of our method and random exploration baselines. As baselines we have uniform random policy (denoted by Random), and a depth-first search policy with random next-step selection (denoted by RandDFS) which allows the agent to backtrack and avoid blindly visiting a node in the current DFS stack. Note that for such DFS, the exploration order of actions is randomized instead of being fixed. Our learned exploration strategy performs significantly better. Fig. 2 shows some example maze exploration trajectories using different approaches.

We conducted an ablation study on the importance of utilizing graph structure, and different variants for modeling the history (see more details in Appendix B.4). We found that 1) exploiting the full graph structure performs significantly better than only using the current node (33%) as the observation or treating all the nodes in a graph as a set (41%) and ignoring the edges; 2) autoregressive aggregation over the history performs significantly better than only using the last step, our best performance is improved by 5% by modeling the full history compared to using a single step observation.

## 4.2 Generating Inputs for Testing Domain Specific Programs

In this section, we study the effectiveness of transferable exploration in the domain of program testing (a.k.a. coverage guided fuzzing). In this setup, our model proposes inputs (test cases) to the program being tested, with the goal of covering as many code branches as possible.

We test our algorithms on two datasets of programs written in two domain specific languages (DSLs), RobustFill [17] and Karel [18]. The RobustFill DSL is a regular expression based string manipulation language, with primitives like `concatenation`, `substring`, *etc.*The Karel DSL is an educational language used to define agents that programmatically explore a grid world. This language is more complex than RobustFill, as it contains conditional statements like `if/then/else` blocks, and loops like `for/while`. Table 1 summarizes the statistics of the two datasets. For Karel, we use the published benchmark dataset[4] with the train/val/test splits;while for RobustFill, the training data was generated using a program synthesizer that is described in [17].Note that for RobustFill, the agent actions are sequences of characters to generate an input string, while for Karel the actions are 2D arrays of characters to generate map layouts (see Fig. 3 for an example program and two generated map layouts), both generated and encoded by RNNs. Training these RNNs for the huge action spaces jointly with the rest of the model using RL is a challenging task in itself.

**Main Results** We compare against two baselines: 1) uniform random policy, and 2) specialized heuristic algorithms designed by a human expert. The objective we optimize is the fraction of unique code branches (for Karel) or regular expressions (for RobustFill) covered (triggered when executing the program on the generated inputs) by the test cases, which is a good indicator of the quality of the generated inputs. Modern fuzzing tools like `AFL` are also coverage-guided.

Fig. 4(a) summarizes the coverage performance of different methods. In RobustFill, our method approaches the human expert level performance, where both of them achieve above $90\%$ coverage. Note that the dataset is generated in a way that the programs are sampled to get the most coverage on human generated inputs[17], so the evaluation is biased towards human expert. Nevertheless, GMETAEXP still gets comparable performance, which is much better than the random fuzzing approach which is widely used in software testing [19]. For Karel programs, GMETAEXP gets significantly better results than even the human expert, as it is much harder for a human expert to develop heuristic algorithms to generate inputs for programs with complex conditional and loop statements.

**Comparing to fuzzers** To compare with fuzzing approaches, we adapted AFL and Neuzz to our problems. We translated all Karel programs into C programs as `afl-gcc` is required in both AFL and Neuzz. We limit the vocabulary and fuzzing strategies to provide guidance in generating valid test cases with AFL. We run AFL for 10 mins for each program, and report coverage using the test cases with distinct execution traces. Note that to get $n$ distinct execution traces AFL or Neuzz may propose $N \gg n$ test cases. Neuzz is set up similarly, but with the output from AFL as initialization.

| | AFL | | | | Neuzz | | | |
|---|---|---|---|---|---|---|---|---|
| # distinct inputs | 2 | 3 | 5 | 10 | 2 | 3 | 5 | 10 |
| joint coverage | 0.63 | 0.67 | 0.76 | 0.81 | 0.64 | 0.69 | 0.74 | 0.77 |
| # inputs tried | 11k | 31k | 82k | 122k | 11k | 14k | 17k | 23k |

Table 3: Karel program coverage with AFL and Neuzz

We report the joint coverage in Table 3. Our approach has a coverage of 0.75 with 1 test case and 0.95 with 5, significantly more efficient than AFL and Neuzz. Correspondingly, when only one test case is allowed, AFL and Neuzz gets coverage of 0.53 and 0.55 respectively, which are about the same as random. Also note that we can directly predict the inputs for new programs (in seconds), rather than taking a long time to just warm up as needed in Neuzz.

However using our approach on standard problems used in fuzzing is challenging. For example, the benchmark from Neuzz consists of only a few programs and this small dataset size makes it difficult to use our learning based approach that focuses on generalization across programs. On the other hand, our approach does not scale to very large scale programs yet. SMT solvers are similar to our approach in this regard as both focus on analyzing smaller functions with complicated logic.

**Comparing to SMT solver** Since it is difficult to design good heuristics manually for Karel, we implemented a symbolic execution [13] baseline that uses the Z3 SMT solver to find inputs that maximize the program coverage. The symbolic execution baseline finds optimal solutions for 412

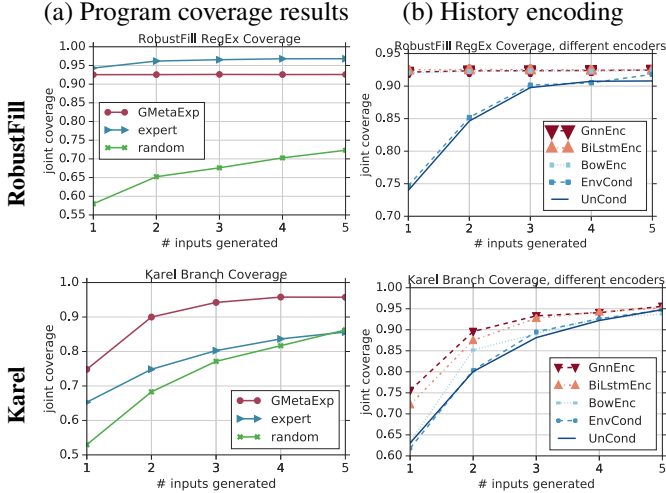

(a) Program coverage results (b) History encoding

Figure 4: Testing DSL programs. (a) Program coverage results. The joint coverage of multiple inputs is reported. (b) Ablation study on different history encoding models.

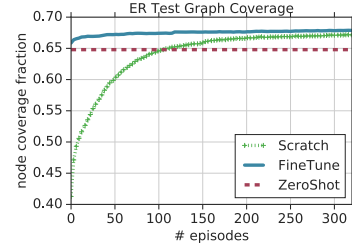

| | Fine-tuning | |
|---|---|---|
| Data | Q-learning | GMETAEXP |
| ER | 0.60 | **0.68** |
| App | 0.58 | **0.61** |
| | Generalization | |
| Data | RandDFS | GMETAEXP |
| ER | 0.52 | **0.65** |
| App | 0.54 | **0.58** |

Table 4: App testing results.

Figure 5: Comparing learning curve of GMETAEXP with different initializations.

out of 467 test programs within the time budget, which takes about 4 hours in total. The average score for the solved 412 cases using a single input is 0.837, which is roughly an 'upper bound' on the single input coverage (not guaranteed to be optimal as we restrict the maximum number of paths to check to 100 and the maximum number of expansions of while loops to 3 to make the solving time tractable). In contrast, GMETAEXP gets 0.76 average score with one input and takes only seconds to run for all the test programs. While the symbolic execution approach achieves higher average coverage, it is slow and often fails to solve cases with highly nested loop structures (i.e., nested repeat or while loops). On the hard cases where the SMT solver failed (i.e., cannot find a solution after checking all top 100 potential paths), our approach still gets 0.698 coverage. This shows that the GMETAEXP achieves a good balance between computation cost and accuracy.

We visualize the test inputs generated by GMETAEXP for two example test programs in Fig. 3 and Fig. 6 in appendix. The covered regular expressions or branches are highlighted. We can observe that the randomly generated inputs can only cover a small fraction of the program. In contrast, our proposed input can trigger many more branches. Moreover, the program also performs interesting manipulations on our generated inputs after execution.

**Effectiveness of program-aware inputs:** When compared with randomly generated program inputs, our learned model does significantly better in coverage. Random generation, however, can trade off efficiency with speed, as generating a random input is very fast. We therefore evaluated random generation with a much higher sample budget, and found that with 10 inputs, random generation can reach a joint coverage (*i.e.*, the union of the coverage of graph nodes/program branches using multiple generated test inputs) of 73%, but the coverage maxed out to only 85% even with 100 inputs (Fig 7 in appendix). This shows the usefulness of our learned model, and the generated program-aware inputs, as we get 93% coverage with just one input.

**Comparison of different conditioning models:** We also study the effectiveness of different exploration history encoders. The encoders we consider here are (1) UnCond, where the policy network knows nothing about the programs or the task, and blindly proposes generally 'good' test cases. Note that it still knows the past test cases it proposed through the autoregressive parameterization of $F(h_t)$ (2) EnvCond, where the policy network is blind to the program but takes the external reward obtained with previous actions into account when generating new actions. This is similar to meta RL [20, 21], where the agent learns an adaptive policy based on the historical interactions with the environment. (3) program-aware models, where the policy network conditions on an encoding of the program. We use BowEnc, BiLstmEnc, and GnnEnc to denote the bag of words encoder, bidirectional LSTM encoder, and graph neural network encoder, respectively.

Fig 4(b) shows the ablation results on different encoders. For RobustFill, since the DSL is relatively simpler, the models conditioned on the program get similar performance; for Karel, we observe that the `GnnEnc` gets best performance, especially when the exploration budget, *i.e.*, the number of inputs is small. One interesting observation is that `UnCond`, which does not rely on the program, also achieves good performance. This shows that, one can find some universally good exploration strategies with RL for these datasets. This is also consistent with the software testing practice, where there are common strategies for testing corner cases, like empty strings, null pointers, *etc.*

## 4.3 App Testing

In this section, we study the exploration and testing problem for mobile apps. Since mobile apps can be very large and the source code is not available for commercial apps, measuring and modeling coverage at the code branch level is very expensive and often impossible. An alternative practice is to measure the number of distinct 'screens' that are covered by test user interactions [22]. Here each 'screen' packs a number of features and UI elements a user can interact with, and testing different interactions on different screens to explore different transitions between screens is a good way to discover bugs and crashes [22, 23].

In this section we explore the screen transition graph for each app with a fixed interaction budget $T = 15$, in the *explore unknown environment* setting. At each step, the agent can choose from a finite set of user interaction actions like `search query, click, scroll`, *etc.* Features of a node may come from an encoding of the visual appearance of the screen, the layout or UI elements visible, or an encoding of the past test logs, *e.g.*, in a continuous testing scenario. More details about the setup and results are included in Appendix B.7.

**Datasets** We scraped a set of apps from the Android app store, and collected 1,000 apps with at most 20 distinct screens as our dataset. We use 5% of them for held-out evaluation. To avoid the expensive interaction with the Android app simulator during learning, we instead used random user inputs to test these apps offline and extracted a screen transition graph for each app. We then built a light-weight offline app simulator that transitions between screens based on the recorded graph. Interacting with this offline simulator is cheap.

In addition to the real world app dataset, we also created a dataset of synthetic apps to further test the capabilities of our approach. We collected randomly sampled Erdős-Rényi (denoted ER in the experiment results) graphs with 15-20 nodes and edge probability 0.1, and used these graphs as the underlying screen transition graph for the synthetic apps. For training we generate random graphs on the fly, and we use 100 held-out graphs for testing the generalization performance.

**Baselines** Besides the RandDFS baselines defined in Sec 4.1, we also evaluate a tabular Q-learning baseline. This Q-learning baseline uses node ID as states and does not model the exploration history. This limitation makes Q-learning impossible to learn the optimal strategy, as the MDP is non-stationary when the state representation only contains the current node ID. Moreover, since this approach is tabular, it does not generalize to new graphs and cannot be used in the generalization setting. We train this baseline on each graph separately for a fixed number of iterations and report the best performance it can reach on those graphs.

**Evaluation setup** We evaluate our algorithms in two scenarios, namely fine-tuning and generalization. In the fine-tuning case, the agent is allowed to interact with the App simulator for as many episodes as needed, and we report the performance of the algorithms after they have been fine-tuned on the apps. Alternatively, this can be thought of as the 'train on some apps, and evaluate on the same set of apps' setting, which is standard for many RL tasks. For the generalization scenario, the agent is asked to get as much reward as possible within one single episode on apps not seen during training. We compare to the tabular Q-learning approach in the first scenario as it is stronger than random exploration; for the second scenario, since the tabular policy is not generalizable, random exploration is used as baseline instead.

**Results** Table 4 summarizes the results for different approaches on different datasets. As we can see, with our modeling of the graph structure and exploration history and learning setup, GMETAEXP performs better in both fine-tuning and generalization experiments compared to Q-learning and random exploration baselines. Furthermore, our zero-shot generalization performance is even better than the fine-tuned performance of tabular Q-learning. This further shows the importance of embedding the structural history when proposing the user inputs for exploration.

We show the learning curves of our model for learning from scratch versus fine-tuning on the 100 test graphs for the synthetic app graphs in Fig 5. For fine-tuning, we initialize with the trained model, and perform reinforcement learning for each individual graph. For learning from scratch, we directly learn on each individual graph separately. We observe that (1) the generalization performance is quite effective in this case, where it achieves performance close to the fine tuned model; (2) learning from the pretrained model is beneficial; it converges faster and converges to a model with slightly better coverage than learning from scratch.

## 5 Related work

Balancing between exploration and exploitation is a fundamental topic in reinforcement learning. To tackle this challenge, many mechanisms have been designed, ranging from simple $\epsilon$-greedy, pseudo-count [1, 3], intrinsic motivation [2], diversity [24], to meta learning approaches that learns the algorithm itself [20, 21], or combining structural noise that address the multi-modality policy distribution [25]. In SLAM literature, the exploration problem is typically known as active SLAM with different uncertainty criteria [26] such as entropy/information based approach [27, 28]. Our work focuses purely on exploring distinct states in graph.

**Exploration for Fuzzing:** Fuzzing explores corner cases in a software, with coverage guided search [23] or learned proposal distributions [29]. To explore the program semantics with input examples, there have been heuristics designed by human expert [17], sampling from manually tuned distributions [30] or greedy approaches [31]. Some recent learning based fuzzing approaches like Learn&Fuzz [29] and DeepFuzz [32] build language models of inputs and sample from it to generate new inputs, but such a paradigm is not directly applicable for program conditional testing. Neuzz [33] builds a smooth surrogate function on top of AFL that allows gradient guided input generation. Rajpal et al. [34] learn a function to predict which bytes might lead to new coverage using supervised learning on previous fuzzing explorations. Different from these approaches that explore a specific task, we learn a transferable exploration strategy, which is encoded in the graph memory based agent that can be directly rolled out in new unseen environments.

**Representation Learning over Structures:** The representation of our external graph memory is built on recent advances in graph representation learning [35]. The graph neural network [9] and the variants have shown superior results in domains including program modeling [15, 8], semi-supervised learning [36], bioinformatics and chemistry [37, 38, 39, 40]. In this paper, we adapt the parameterization from Li et al. [15], the graph sequence modeling [41], and also the attention based [42] read-out for the graph.

**Optimization over Graphs:** Existing papers have studied the path finding problems in graph. The DOM-Q-NET [43] navigates HTML page and finishes certain tasks, while Mirowski et al. [44] learns to handle complex visual sensory inputs. Our task is seeking for optimal traversal tour, which is essentially NP-hard. Our work is also closely related to the recent advances in combinatorial optimization over graph structured data. The graph neural network can be learned with one-bit [45] or full supervision [46, 47] and generalize to new combinatorial optimization problem instances. In the case that lacks supervision, the reinforcement learning are adapted [48]. Khalil et al. [49] uses finite horizon DQN to learn the action policy. Our work mainly differs in two ways: 1) the full structure of graph is not always observed and instead needs be explored by the agent; 2) we model the exploration history as a sequence of evolving graphs, rather than learning Q-function of a single graph.

## 6 Conclusion

In this paper, we study the problem of transferable graph exploration. We propose to use a sequence of graph structured external memory to encode the exploration history. By encoding the graph structure with GNN, we can also obtain transferable history memory representations. We demonstrate our method on domains including synthetic 2D maze exploration and real world program and app testing, and show comparable or better performance than human engineered methods. Future work includes scaling up the graph external memory to handle large software or code base.

**Acknowledgments**

We would like to thank Hengxiang Hu, Shu-Wei Cheng and other members in the team for providing data and engineering suggestions. We also want to thank Arthur Guez, Georg Ostrovski, Jonathan Uesato, Tejas Kulkarni, and anonymous reviewers for providing constructive feedbacks.

## Footnotes

*Work done during an internship at DeepMind, when Hanjun was in Georgia Institute of Technology

[2] When it is unknown, we can simply divide the reward by $T$ to normalize the total reward.

[3] http://lcamtuf.coredump.cx/afl/

[4]https://msr-redmond.github.io/karel-dataset/

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
