[Supplementary Material]

# A  Statement of the problem

## A.1  App covering is at least as hard as finding Hamiltonian Path

Suppose we already know all the nodes (screens) in the app graph, then the app covering/exploration problem can be abstracted as: *given a graph with $N$ nodes, what's the maximum number of nodes one can visit by traversing the graph with at most $T$ steps.*

To show the NP-completeness, we first convert the optimality problem to the decision problem: *given a graph with $N$ nodes and a number $M$, where there exists a traversal plan that can visit at least $M$ nodes within $T$ steps.*

Hamiltonian Path $\rightarrow$ App Covering: given a Hamiltonian Path problem instance with $n$ nodes, we set $N = n$, $M = n$ and $T = n$ in App Covering, then solves the App Covering automatically solves Hamiltonian Path.

Note that the above argument is based on the assumption that we already know the entire graph. However our task is to explore the graph, in which we don't know the full graph yet. This makes the problem even harder.

## A.2  Graph optimal traversal order

If the graph is a tree, then finding the optimal traversal order reduces to dynamic programming. The best strategy is to find the longest path starting from the beginning node, and explore this path in the end. In this way, once all the nodes are explored, the longest path doesn't need the backtracking, thus saving the exploration budget. If it is a general graph, then it reduces to the problem mentioned in Appendix A.1. Then in both cases, the optimal graph traversal order is not trivial.

# B  Experiment Details

## B.1  Model architectures

To parameterize the Eq Eq. (4), we use the following realization:

$$
\begin{aligned}
m_v^{(l+1)} &= \text{Aggregate}(\{\text{MLP}[\mu_v^{(l)}, \mu_u^{(l)}, k]\}_{u \in \mathcal{N}^k(v)}, \\
& \quad k \in 1, 2, \ldots, K) \qquad (5) \\
\mu_v^{(l+1)} &= \text{GRU}(\mu_v^{(l)}, m_v^{(l+1)}) \qquad (6)
\end{aligned}
$$

Here MLP$(\cdot)$ is a fully connected network, and the aggregation method is tuned within $\{sum, mean, max\}$. Finally, following Li et al. [15], we use GRU as gating function for the node embedding update. At the beginning, $\mu_v^{(0)}$ is set to be the plain node feature, such as tokens in program, or screen features in app exploration.

To get the representation of entire graph memory, we can simply do the summation over all node embeddings, *i.e.*, $g(G, c) = \sum_v \mu_v^{(L)}$ where $L$ is the number of message-passing steps used in GGNN. However, we found the attentive aggregation obtains best empirical performance:

$$
\alpha_v = \frac{\exp(W^\top \mu_v)}{\sum_{u \in G} \exp(W^\top \mu_u)}, \qquad (7)
$$

$$
g(G, c) = \sum_{v \in G} \alpha_v \mu_v^{(L)} \qquad (8)
$$

To embed the history $F(h_t)$, the simplest way is to use the graph memory read-out $g(G_t, c_t)$, since in many cases the current status of the graph is sufficient to tell the history. A stronger parameterization is to use auto-regressive model, where $F(h_t) = \text{LSTM}(F(h_{t-1}), g(G_t, c_t))$ and let $F(h_1) = g(G_0, c_0)$.

In most experiments, we use $L = 5$ for this graph memory read-out function.

## B.2 Coverage metric for programs

The RobustFill language defines programs as concatenation of regular-expression based substring operations. For example, consider the program Concat(e1,e2), where e1 ≡ Substr((Num,3,End),(Proper,2,Start)) and e2 ≡ Substr((';',1,End),(',',4,Start)). This program concatenates two input substrings: i) substring between $3^{rd}$ number and $2^{nd}$ propercase, and ii) substring between $1^{st}$ ';' and $4^{th}$ ','. In order to trigger these regular expressions, the input string should consists of at least 3 numbers, 2 propercase tokens, $1^{st}$ ';', and $4^{th}$ ',' tokens. Karel language, on the other hand, uses control statements such as if/else and while loops. To maximize the coverage, the inputs should be able to reach both `True/False` conditions of each control statement. In both RobustFill and Karel experiments, we measure the coverage as the fraction of statements reached. For Karel, we measure the branch coverage where we check if the inputs can reach both blocks of a conditional, whereas for RobustFill, we measure the number of regular expression based substring expression successfully triggering to generate a non-empty result (without an exception). The coverage is normalized to $[0, 1]$, where 1 means the full coverage.

## B.3 Details about baseline algorithms for program coverage

**Details about human expert design** For RobustFill, the human expert generated the test inputs and programs in a different way. In brief, the test inputs are generated first that induces an over-approximation of possible program expressions, and then the programs are sampled in a way that almost every regular expression can be triggered. Ideally this procedure would get 100% coverage for the program, but it is challenging to handle the conflicts between different regular expression requirements and the resulting position indices when evaluated on concrete inputs while also maintaining a maximum output length constraint. Finally this method gets 96% coverage when 10 inputs are generated for each program. Note that for our proposed GMETAEXPtechnique, we are asked to generate inputs based on the program, which is harder than the human expert's setting. Nevertheless, we still get $\sim 93\%$ coverage, which is close to human's performance.

For Karel, it is much harder to design the heuristics manually because of the complexity of the DSL. We noticed that quite often the randomly generated inputs lead to runtime errors in a Karel program. These errors include infinite loops, or invalid actions in certain situations. The execution terminates once an error is encountered. Thus the heuristic method being used here is to guarantee the successful execution of the program.

**Finding inputs for Karel program using symbolic execution** We also implemented a symbolic execution baseline using the Z3 SMT solver to find inputs that maximize coverage of Karel programs. In this baseline, we first represent the input maze symbolically, where each grid in a maze is represented as a symbolic integer $s_{ij} \in \{-1, 0, 1\}$ representing either the grid $i, j$ in the maze is a wall, an empty slot, or a slot with one marker. Similarly, we represent the hero as a symbolic tuple $(h_x, h_y, f)$, where $h_x \in [0, |row|], h_y \in [0, |col|]$ representing the initial position of the hero and $f \in \{0, 1, 2, 3\}$ representing the facing of the hero. With this representation, we extract all paths of program, rank them based on the number of branches they covered (i.e., coverage score), and finally use Z3 to check whether there exists an input that follows the path. If a solution is found, then the values of the symbolic values represent the configuration of the maze and the hero, which is also an optimal input for program coverage; otherwise we continue to examine the next path, until finding a satisfying input or timeout (after failing on all of the top 100 paths in our experiment). During the path extraction process, whenever we encounter a condition expression (i.e., *If*, *IfElse*, *While*), we split the path into two. Since *while*-loop can be infinitely expanded, we only consider a finite expansion (unrolling at most 3 times for each loop in our experiment). As the number of total paths is exponential in the number of conditions and examine all possible paths can be prohibitively expensive, we restrict the baseline to only examine the top ranked 100 paths for a given program. This restriction prevents the baseline to check all paths for satisfiable solutions when the program contains a large number of paths but allows the experiment to finish with 4 hours. As a result, the baseline can fail in finding inputs for some examples (failed 55 out of 467 test cases).

## B.4 Ablation studies on history conditioning models

Here we show GMETAEXP with different history conditioning method, and report the exploration coverage for 2D maze task in Table 5 and Table 6. These two tables use different configurations of

| Policy Net | History | Test Performance |
|---|---|---|
| Random | - | 0.329 |
| Random DFS | - | 0.539 |
| Node | Non-Autoregressive | 0.3363 |
| Node | Autoregressive | 0.6603 |
| Pool | Non-Autoregressive | 0.4151 |
| Pool | Autoregressive | 0.6671 |
| Graph | Non-Autoregressive | 0.6634 |
| Graph | Autoregressive | 0.7160 |

Table 5: Maze results, 1 layer GNN feature

| Policy Net | History | Test Performance |
|---|---|---|
| Random | - | 0.329 |
| Random DFS | - | 0.539 |
| Node | Non-Autoregressive | 0.3409 |
| Node | Autoregressive | 0.7063 |
| Pool | Non-Autoregressive | 0.5217 |
| Pool | Autoregressive | 0.7074 |
| Graph | Non-Autoregressive | 0.6780 |
| Graph | Autoregressive | 0.7340 |

Table 6: Maze results, 2 layer GNN feature

unsupervised graph neural network features (see Sec. B.6). Basically the more layers used in this feature extractor, the larger receptive field the agent can have in the maze.

We have three different history encoding models, namely 'Node', 'Pool' and 'Graph'. Here 'Node' only encodes the information of current location in the maze, while 'Pool' encodes all the visited nodes with an unordered pooling (*i.e.*, ignores the structural information). The 'Graph' conditioning model instead uses all the information in the graph external memory. We can see if we use non-autoregressive policy network, then the policy with 'Graph' conditioning outperforms the other two significantly. While using autoregressive model, basically all three method should have encoded same amount of information. Nevertheless, the 'Graph' conditioning model still performs better than the others, especially when the node feature quality is low (*e.g.*, in Table 5). This is mainly because of the forgetting issue in LSTM, where the past history is not effectively captured. This shows that encoding the graph history in this way can properly remember the exploration history, thus being more effective in the exploration task. Also it suggests that, the history includes information about how the graph and coverage evolves and how we get to the current state, which might be more beneficial than just a current snapshot of graph state.

## B.5 Ablation studies on program testing

Here we compare the effectiveness of learned exploration strategy and random exploration strategy on RobustFill dataset. As is shown in Figure 7, 100 randomly generated inputs for the program can jointly get 85% coverage on held-out test set, while the learned GMETAEXP gets 92% within at most 5 generated inputs on the test set in zero-shot setting. This shows the significance of the learned exploration strategy. Figure 6 shows an example of effective coverage for the program. In this example, the proposed test case gets full coverage of the program, but the randomly generated one only gets 0.36.

**The importance of the run-time information:** The run-time information $c_t(\cdot)$ we obtained from executing the program can be important for generating meaningful test cases. Figure 4(c) shows the performance of the program-aware `GnnEnc` model with and without run-time information $c_t$. In RobustFill having access to $c_t$ made a big difference, while in Karel the difference is much smaller.

- Random Input

  `F:ua.q3'2s:PhTG,IqIz`

  Output:

  `3'2s,`

  Coverage: **0.36**

- MetaExp Input

  `'Ym71zA9Nm2XmXAf""""`

  Output:

  `zA9Nm2XmXAf"A9Nm","`

  Coverage: **1.0**

```
StrConcat(
  SubStr
    RegPos
    ConstTok(", 2, Start)
    RegPos
    RegexTok([a-z]+, 2, Start)
  SubStr
    RegPos
    RegexTok([a-z]+, 3, End)
    RegPos
    RegexTok([a-z]+, 2, End)
  SubStr
    -1
    RegPos
    ConstTok(", 4, Start)
  ','
  SubStr
    RegPos
    ConstTok(", 1, End)
    RegPos
    ConstTok(", 2, End)
)
```

Figure 6: Example test cases generated for RobustFill.

Figure 7: RegEx coverage comparison on RobustFill.

(a) RobustFill    (b) Karel

Figure 8: Ablation on with/without history run-time information.

## B.6 Unsupervised Graph Neural Network for Representation Learning

To have generalizable representation of plain graphs (*i.e.*, graphs with no node or edge features), we use the unsupervised link prediction objective to train a graph neural network (GNN) and thus obtain the plain graph features.

Specifically, given a graph $G = (V, E)$ with node set $V$ and edge set $E \subseteq V \times V$, we try to associate each node with an embedding vector $\mu_v \in \mathbb{R}^d, \forall v \in V$, so as to decompose the adjacency matrix $A \in \mathbb{R}^{|V| \times |V|}$, i.e.,

$$\min_{\{\mu_v\}_{v \in V}, W \in \mathbb{R}^{d \times d}} \sum_{i \in V} \sum_{j \in V} (A_{i,j} - \mu_i^\top W \mu_j)^2 \tag{9}$$

These embedding vectors thus capture the structural information from the graph. To enable the generalization of the features, we cannot directly optimize the vectors for each graph, but instead, we will parameterize the embedding using GNN:

$$\mu_v^{(L+1)} = f(\mu_v^{(L)}, \{\mu_u\}_{u \in \mathcal{N}(v)}) \tag{10}$$

and let $\mu_v = \mu_v^{(L+1)}$ be the last outcome of this iterative embedding process. We use GGNN to parameterize $f$. Since we don't have node features, we simply assign a constant to $\mu_v^{(0)} = 0$. Note that the $L$ denotes the receptive field for each vector. To enable the zero-shot generalization, sometimes one can only observe the local graph, thus we need to have small $L$ in these cases. Overall, the objective is written as:

$$\min_{f, W \in \mathbb{R}^{d \times d}} \mathbb{E}_{G=(V^{(G)}, E^{(G)}) \in \mathcal{G}} \|A^{(G)} - \mu^{(G)\top} W \mu^{(G)}\| \tag{11}$$

where $\mu^{(G)}$ is the embedding of graph $G$ computed by embedding function $f$.

Figure 9: Example App graphs for exploration experiment. Graphs in the top row are random synthetic ones, while graphs in the bottom row are collected from real-world Android Apps. Each node represent a screen, where the red node is the start screen.

Figure 10: Exploration on synthetic apps starting from the blue node. The darkness of node colors indicates the visit count of each node. The white nodes are not visited.

## B.7 Building offline App simulator for Reinforcement Learning

We collect 1,000 small apps from the android app store for this experiment. Instead of interacting with the mobile app simulator during reinforcement learning, we collect the exploration histories from the app. The historical data is a list of tuples in the form $sID_i, tID_i, act_{i_i}$, namely from screen with $sID_i$ to screen $tID_i$ using action $act_i$. Using these tuples we can rebuild the transition graph for the app. Now we can treat the app as a graph $\mathcal{G}$, and treat it as graph exploration problem. Fig 9 contains the example graph representation for both synthetic and real-world apps.

For synthetic apps, we also visualize the exploration history in Figure. 10. The darkness on nodes represent the corresponding visit count. The darker the more visits. We can see the random exploration gets stuck in some local neighborhood, while our algorithm explores better with less repetitions on the visited nodes.

Since it is a 'fake' app built from the transition tuples, there won't be bugs. Thus the purpose is to demonstrate the effectiveness of exploration. Also, since in this case no interaction with the mobile app simulator is needed during reinforcement learning, we can quickly test different models and experiment setups.

In the real-world setting, we can still adopt this approach to speed up the testing. Since the testing is continual which can last for several years for a single app (*e.g.*, testing with different versions of the same app continually), we can collect the logs along with the testing procedure, and use these offline logs to train the RL agent.