[Reviews · NeurIPS 2019]

Reviewer 1



Contributions: The paper uses a combination of GNN and RL techniques to learn to explore graph-structured environments. The method is used in a synthetic maze exploration experiment, and for program testing to improve test coverage. Key contributions are: Framing of the graph exploration problem using a model that uses (1) a graph memory representing the visited state, (2) incorporating the exploration history in the prediction of the next action, and (3) different weights per time step. Framing program testing (including app testing) as a graph-exploration problem. Ablation studies exploring the overall importance of weight sharing (per RL time step), graph structure, exploration history, and different methods of encoding the program in the program testing. Comments: Originality: There are prior works where GNN + RL are combined that has a similar flavor (e.g. where the states are represented as a GNN, or where actions are represented by nodes in a GNN). This work combines those ideas with the goal of learning to explore. In particular, I don’t recall seeing the idea of using an evolving graph memory. https://openreview.net/pdf?id=HJgd1nAqFX https://arxiv.org/pdf/1906.05370.pdf https://graphreason.github.io/papers/7.pdf Clarity: The paper is well-written and easy to follow. I’m curious about: - In Table 2 and Figure 2, where the “randomness” in the “random next-step selection” in RandDFS comes from. In particular, how much worse is RandDFS compared to actual DFS? - In Figure 2, it would be useful & interesting to see the actual trajectory taken by the various agents, or at least the initial position of the agents. - In Figure 4, I’m not sure what “# inputs generated” means. (At first I thought that means T = 5 in these experiments, but that doesn’t really make sense?) - Also in figure 4, what does the y-axis label “joint coverage” mean? Quality: The submission appears technically sound, with experiments that are relatively different from each other. The maze exploration task with the ablation study shows the performance gain in using full graph structure and using the overall history. Reporting the standard deviation over the 100 held-out mazes would be helpful in analyzing the proposed approach. That using the overall history performs better is a bit unintuitive to me. The program testing tasks also have ablation studies that show how the encoding of the program being tested affect coverage. However, it is unclear whether using the overall history would be beneficial at all. The author also does not compare against any fuzzing or neural fuzzing approaches. The app-testing task shares similarities with the maze task, but in a more realistic scenario. It would be helpful to provide an intuition about what a real app traversal graph looks like (not the generated ER graph) for comparison against the maze task.

Reviewer 2



I appreciate the contribution of introducing the new problem framework of graph exploration. The proposed GNN-based RL algorithm is also well designed and reasonable. My major concerns are as follows. 1) It is unclear why the proposed framework is better for specific tasks and whether this graph exploration framework, in general, is expressive enough to adapt to task-specific characteristics. Initially, when I’ve seen the title with the term “transferable”, I’ve expected that the graph exploration is transferable between different `tasks’. However, the authors use the term “transferable” in the sense that their trained agent works for `unseen’ graphs in the same task. In this sense, I think many task-specific methods in the literature would also be transferable. If so, I am not sure what is the advantage of the proposed graph exploration network than a task-specific method. E.g., does the other maze exploration or fuzzing methods generalize to unseen data worse than the proposed one? I think this needs to be clearly justified. 2) The baselines are too simple and no comparison with any state-of-the-art method for each task is provided. For maze exploration, there are many recent methods, (e.g., Mirowski et al., Learning to navigate in complex environments, ICLR 2017), but this paper does not compare with them. For fuzzing, the paper only compares with random strategy and an expert-designed heuristics even though there are many methods for fuzzing (e.g, Godefroid et al, Learn&Fuzz: Machine Learning for Input Fuzzing, AES 2017, and Liu et al, DeepFuzz: Automatic Generation of Syntax Valid C Programs for Fuzz Testing, AAAI 2019). 3) Minor comments - For equation (4), if you use the padding of one extra bit c_t(v), it would be more helpful to add this term explicitly. - How do you initialize node embedding for program testing and maze exploration? Is it done in the same way for each dataset? - In maze exploration, it would be more informative to show the detailed trajectory of each agent so that we understand how the network explores the maze.

Reviewer 3



The idea of this article is new and innovative. It combines machine learning with the exploration of spatial data, which has a lot of application scenarios. The author also gives a specific application scenario, software testing of domain-specific programs and mobile applications.

Reviewer 4



Although the problem of exploration in a graph-structured space seems to be a pretty natural extension of the exploration problem initially proposed in other contexts, e.g., game playing environment, their proposed problem setup is not well-studied yet, thus it is a meaningful topic. However, the novelty of their approach is very minor. The ideas of RL + GNN and RL for exploration arenot new, as stated in their related work. The idea of maintaining a graph-structured memory is not new; e.g., [1] maintains the evolution of the graph structure over time. In particular, RL + GNN + exploration is not completely novel as well; [2] studies the problem of goal-oriented web navigation, using RL + GNN. Although their formulation is different from the exploration task, they are actually solving a harder problem where the reward could be very sparse. Regarding the experiments, although the latter two tasks, i.e., program testing and app testing, are real-world tasks, the datasets are still more synthetic than the benchmarks used in previous work on related applications. For example, FlashFill and Karel have simplified grammars that constrain the search space for fuzzing. Why not evaluate on standard fuzzing benchmarks as in [3], and compare with existing neural, RL, and classific fuzzing approaches (you can find comprehensive baselines in [3])? If the authors can demonstrate superior performance on these more realistic benchmarks, the results could be much more convincing. [1] Daniel D. Johnson, Learning Graphical State Transitions, ICLR 2017. [2] Jia et al., DOM-Q-NET: Grounded RL on Structured Language, ICLR 2019. [3] She et al., NEUZZ: Efficient Fuzzing with Neural Program Smoothing, IEEE S&P 2019.

[Author Response · NeurIPS 2019]

We thank the reviewers for the reviews and suggestions, which we will incorporate into the next revised version. For brevity we denote the reviewers by [R1], [R2], [R3], [R4] in the detailed responses below.

[R2][R4] **Comparison with other fuzzing techniques.** Our graph exploration approach is designed for tasks where we want to maximize exploration with a small interaction (number of inputs) budget. The small budget, in particular, challenges our model to learn to reason about the environment or source code. The fuzzing approaches, on the other hand, trades off test case quality with speed and scale, by using randomly (cheaply) generated test cases, and they are more suitable for cases where the exploration budget is not a limiting factor and programs do not require complex reasoning. Another significant diff. between our approach and fuzzing is that our model learns about the distribution of programs (hence we need a large dataset of programs to train on), and therefore can transfer and perform better on unseen programs; while fuzzing approaches start from scratch for each program without any cross-program adaptation.

To compare our approach with fuzzing, we adapted AFL [Zalewski] and Neuzz [She et al., 2019] to our problems. We translate all Karel programs into C programs as `afl-gcc` is required in both AFL and Neuzz. We limit the vocabulary and fuzzing strategies to provide guidance in generating valid test cases with AFL. We run AFL for 10 mins for each program, and report coverage using the test cases with distinct execution traces. Note that to get $n$ distinct traces AFL or Neuzz may propose $N \gg n$ test cases. Neuzz is set up similarly, and initialized with the output from AFL.

| # distinct inputs | 2 | 3 | 5 | 10 | | 2 | 3 | 5 | 10 | | | |
|---|---|---|---|---|---|---|---|---|---|---|---|---|
| joint coverage | 0.63 | 0.67 | 0.76 | 0.81 | | 0.64 | 0.69 | 0.74 | 0.77 | | Ours | 0.75 |
| # inputs tried | 11k | 31k | 82k | 122k | | 11k | 14k | 17k | 23k | | AFL | 0.53 |
| | | | | | | | | | | | Neuzz | 0.50 |

Table 1: Karel Coverage with AFL     Table 2: Karel Coverage with Neuzz     Table 3: Single input coverage

We report the joint/single coverage with best configs above. For comparison, our approach has a coverage of 0.75 with 1 test case and 0.95 with 5, significantly more efficient than AFL (0.76) and Neuzz (0.74). Also note that we can directly predict the inputs for new programs (in seconds), rather than taking a long time to just warm up as needed in AFL.

Using our approach on standard problems used in fuzzing is challenging. For example, the benchmark from Neuzz consists of only a few programs and its size makes it difficult to use our learning based approach that focuses on generalization across programs. On the other hand, our approach does not scale to very large scale programs yet and scalability is a challenge for future work. SMT solvers are similar to our approach in this regard as both focus on analyzing smaller functions with complex logic.

[R1][R2][R4] **More related work in RL and fuzzing literature.** We thank the reviewers for pointing out relevant papers, which we will properly cite in our revision. Here we briefly summarize and discuss the connections and differences to prior work. In GNN/RL related work, DOM-Q-NET [Jia et al., 2019] navigates HTML page and finishes certain tasks while the "Learning to navigate" paper [Mirowski et al., 2017] is more about handling complex visual sensory inputs. These two are like path finding, while our task is seeking optimal traversal tour (which is NP-hard). Also these two do not model the history. The Neural Graph Evolution [Wang et al., 2019] searches for the best graph structure, while we are learning to explore on a graph. The Graphical State Transitions paper [Johnson, 2017] builds a graph structure for supervised learning tasks like Q&A. Adapting them to an RL problem alone is highly non-trivial and requires significant work. For Fuzzing, we compared against an SMT solver and human experts, which are *not* weak baselines [R2]. The related work of Learn&Fuzz [Godefroid et al., 2017] and DeepFuzz [Liu et al., 2019] learn language models of inputs (in an unsupervised way) and sample from it to generate new inputs, which is not useful in our setting as we condition the input generation on the program being tested. AFL and Neuzz are more appropriate baselines and we report results comparing with them for testing Karel programs above.

[R1] **"randomness" in RandDFS:** At each state in DFS, the RandDFS picks the next unexplored action at random; while the standard DFS picks an action according to some fixed order. In random environments, these two should perform similarly. [R1] **what "# inputs generated" means:** This is the number of actions (input test cases) the agent used to explore the environment / cover the program. Note that each action corresponds to the entire generated input structure (e.g., entire maze, or string). Reward is given after generating each such input structure. [R1] **"joint coverage" in Fig 4:** It is the union of the coverage of graph nodes (program branches) using multiple generated test inputs. [R1] **using history performs better is unintuitive:** The history includes information about how the graph and coverage evolves and how we get to the current state, which might be beneficial. [R1] **what a real app traversal graph looks like:** We have provided the graphs in Fig 9 in Appendix. [R2] **transfer across tasks:** By "zero-shot" and "transfer" we mean once learned our model can be applied directly to unseen graphs (i.e. tasks) without changing the model parameters. The learning based approaches in the literature are typically trained on and only works for a single task (or program); while the fuzzing approaches do not learn and cannot adapt as effectively as our approach (see Table 1, 2 above). [R2] **"How do you initialize node embedding"** Nodes are initialized with corresponding tokens in the syntax tree for Karel/RobustFill; and constant vectors for maze exploration, as there's no features associated with them. [R3] Thank you for the overall positive recognition of our work. [R4] Thanks for the constructive feedback. We will include more discussion of related work, and the comparison with AFL and Neuzz reported above in the next revision.

[Meta-Review · NeurIPS 2019]

The paper is concerned with learning a general exploration policy, trained using reinforcement learning and considering a distribution of graph-structured environments. A motivating application is coverage-guided program testing (fuzzing). The paper generated quite some discussion among the reviewers, concerning the novelty compared to recent works, and the relevance of the contribution. The rebuttal did a very good job in clarifying the novelty issue, making it clear that the goal is similar to a touring problem (visiting all states, i.e. branches in the program graph, based on graph exploration history) as opposed to path finding or graph design. The additional results (comparison with AFL and Neuzz required by the reviewers, in the rebuttal) are also found to be very convincing regarding the efficiency of the approach in the small budget (in terms of input generation) regime. The comparison with expert and SMT solvers (section 4.2) gives a sufficient evidence that the paper brings a contribution at the state of the art in an important niche, the fuzzing of small scale programs. The comparison of the different encoding of the programs (bag of words, BiLSTM and GraphNN) is also very interesting. For all these reasons, the Area Chair thinks the paper can be accepted. In the camera-ready version, the authors will do their best to situate their contribution w.r.t. the application domains, make it clear that GNN can be trained to perform smart exploration, and give precisions on the scalability of the approach.